# Molecular Epidemiology of *Mycobacterium tuberculosis* Complex Strains in Urban and Slum Settings of Nairobi, Kenya

**DOI:** 10.3390/genes13030475

**Published:** 2022-03-08

**Authors:** Glennah Kerubo, Perpetua Ndungu, Yassir Adam Shuaib, Evans Amukoye, Gunturu Revathi, Susanne Homolka, Samuel Kariuki, Matthias Merker, Stefan Niemann

**Affiliations:** 1Molecular and Experimental Mycobacteriology, Research Center Borstel, 23845 Borstel, Germany; vet.aboamar@gmail.com (Y.A.S.); shomolka@fz-borstel.de (S.H.); mmerker@fz-borstel.de (M.M.); 2School of Medicine, Kenyatta University, Nairobi 43844-00100, Kenya; 3Jomo Kenyatta University of Agriculture and Technology, Nairobi 62000-00200, Kenya; ndunguperpetual@gmail.com; 4College of Veterinary Medicine, Sudan University of Science and Technology, Khartoum North 13321, Sudan; 5Centre for Respiratory Disease Research, Kenya Medical Research Institute, Nairobi 54840-00200, Kenya; amukoye@gmail.com; 6Aga Khan University Hospital, Nairobi 30270-00100, Kenya; gunturu.revathi@aku.edu; 7Centre for Microbiology Research, Kenya Medical Research Institute, Nairobi 54840-00200, Kenya; samkariuki2@gmail.com; 8German Center for Infection Research (DZIF), Partner Site Hamburg-Lübeck-Borstel-Riems, 23845 Borstel, Germany; 9Evolution of the Resistome, Research Center Borstel, 23845 Borstel, Germany

**Keywords:** tuberculosis, *Mycobacterium tuberculosis*, whole-genome sequencing, molecular epidemiology, Nairobi

## Abstract

Kenya is a country with a high tuberculosis (TB) burden. However, knowledge on the genetic diversity of *Mycobacterium tuberculosis* complex (MTBC) strains and their transmission dynamics is sparsely available. Hence, we used whole-genome sequencing (WGS) to depict the genetic diversity, molecular markers of drug resistance, and possible transmission clusters among MTBC strains in urban and slum settings of Nairobi. We analyzed 385 clinical MTBC isolates collected between 2010 and 2015 in combination with patients’ demographics. We showed that the MTBC population mainly comprises strains of four lineages (L1–L4). The two dominating lineages were L4 with 55.8% (*n* = 215) and L3 with 25.7% (*n* = 99) of all strains, respectively. Genome-based cluster analysis showed that 30.4% (117/385) of the strains were clustered using a ≤5 single-nucleotide polymorphism (SNP) threshold as a surrogate marker for direct patient-to-patient MTBC transmission. Moreover, 5.2% (20/385) of the strains were multidrug-resistant (MDR), and 50.0% (*n* = 10) were part of a genome-based cluster (i.e., direct MDR MTBC transmission). Notably, 30.0% (6/20) of the MDR strains were resistant to all first-line drugs and are part of one molecular cluster. Moreover, TB patients in urban living setting had 3.8 times the odds of being infected with a drug-resistant strain as compared to patients from slums (*p*-value = 0.002). Our results show that L4 strains are the main causative agent of TB in Nairobi and MDR strain transmission is an emerging concern in urban settings. This emphasizes the need for more focused infection control measures and contact tracing of patients with MDR TB to break the transmission chains.

## 1. Introduction

Tuberculosis (TB) is still a major cause of morbidity and mortality worldwide, especially in Sub-Saharan Africa [1,2,3]. In 2020, TB was responsible for 9.9 million new cases, of which 25% were in the World Health organization (WHO) African region [1]. Kenya is among the highest TB burden countries with an incidence of 259/100,000 population, which is twice as much as the global average [1,2,3]. The total number of new and relapse TB case notifications was 71,646 [1]. The HIV/AIDS epidemic and the emergence of drug-resistant *Mycobacterium tuberculosis* complex (MTBC) strains complicated the effective control of TB in the country [4,5]. Additionally, the emergence of SARS-CoV-2 is expected to have a long-lasting impact leading to an increase in the burden of TB in Kenya due to the disruption of TB health care services in 2020 [6,7]. Moreover, the lack of resources to carry out proper and definitive diagnosis of TB fuels the country’s TB epidemic as TB diagnosis is solely performed using smear microscopy in Kenya. This technique has a variable sensitivity and does not allow for resistance prediction or differentiation between infections due to MTBC and non-tuberculous mycobacteria (NTM) [8,9]. TB cases are therefore treated using empirical treatment which may foster the development and transmission of resistant MTBC strains by long periods of infectivity [10].

In Kenya, knowledge on genetic drug resistance markers of MTBC strains is inadequate, especially in potential transmission hotspots, such as the capital Nairobi. Molecular epidemiological studies have suggested that transmission is a major driver of drug susceptible and drug resistant TB epidemics, not only in Eastern Europe but also in Africa [10,11,12,13]. Indeed, MTBC transmission is one of the major constraints that hinder TB control, particularly in poor settings (e.g., slums) where the risk of transmission is high due to overcrowding [1,13]. Evidence has disclosed high TB prevalence in slums in Kenya [14] and other African countries, such as South Africa [15,16]. Furthermore, patients in slums may have limited access to TB treatment centers.

Here, we collated MTBC strains from two patient cohorts in Nairobi to analyze differences in the genetic diversity, transmission dynamics, and molecular drug resistance profiles of MTBC strains circulating in slum and urban settings.

## 2. Materials and Methods

### 2.1. Study Design and Setting

We enrolled self-presented presumptive TB patients in this cross-sectional study who had respiratory symptoms and were smear-positive from 13 randomly selected health centers located in Nairobi and its surroundings (Appendix A). The first cohort (*n* = 204) was enrolled between May 2010 and May 2013, and the second cohort (*n* = 275) was enrolled from September 2014 to May 2015. Three of the health care centers selected are TB referral centers (Appendix A), of which one center is located in slums (i.e., highly populated or informal settlements with weak building quality, inadequate housing or very small living spaces, and squalid conditions which are often associated with poverty) and serves up to 200 patients per week. Each patient was required to give a spot (i.e., collected at the time of first visit of the patient to clinic/laboratory) and an early morning sputum sample for analysis. Both spot and early morning samples were pooled and used for microbiological analysis. Demographic data were gathered using a structured questionnaire, and each recruited patient was assigned a unique personal identifier that was used throughout the study period.

### 2.2. Laboratory Procedures

#### 2.2.1. Sputum Decontamination, Mycobacterial Culture, and DNA Extraction

All specimens were processed and decontaminated following the standard *N*-acetyl-l-cysteine-–sodium hydroxide (NALC-NaOH) method for digestion, decontamination, and concentration [17]. The sediment was resuspended in about 1 to 2 mL phosphate buffer (pH 6.8) and mixed thoroughly. A smear was prepared for acid-fast staining [9,18].

Mycobacterial cultures were performed by adding 0.25 mL of the suspended decontaminated samples into the BACTEC MGIT 960 (Becton Dickinson, Franklin Lakes, NJ, USA) and onto solid Löwenstein–Jensen (LJ) culture media [19].

DNA extraction from mycobacterial colonies was carried out using the cetyl trimethylammonium bromide (CTAB) method as described by Somerville et al. [20]. The DNA was reconstituted with molecular grade water and stored at −20 °C until it was sent to the Molecular and Experimental Mycobacteriology Laboratory, Research Centre Borstel in Germany, where molecular characterization of the MTBC strains was conducted.

#### 2.2.2. Whole-Genome Sequencing 

DNA libraries for whole-genome sequencing (WGS) were prepared with the Nextera (XT) kit from Illumina (San Diego, USA) according to the manufacturer’s instructions [21]. Pooled DNA libraries were then loaded into NextSeq Reagent cartridges for sequencing on a NextSeq system (Illumina, San Diego, CA, USA). Resulting sequencing reads were submitted to the European Nucleotide Archive under the project accession number PRJEB50767 and subsequently mapped to the H37Rv reference genome (GenBank ID: NC_000962.3) by Burrows–Wheeler alignment (BWA) tool aiming for a minimum of 50-fold average genome-wide coverage [22,23]. We considered single nucleotide polymorphisms (SNPs) with at least 4 reads in both forward and reverse orientation, 4 reads calling the allele with at least a Phred score of 30, and 75% allele frequency for a concatenated sequence alignment. SNP positions that had reliable base call (as described above) in at least 95% of the strains were concatenated to a sequence alignment. SNPs from repetitive regions were excluded, including those which occurred within a window of 12 base pairs in neighboring strains [23].

A web tool was used for detecting isolates that harbored more than one phylogenetic lineage (i.e., mixed infections or laboratory contaminations) [24]. These samples/isolates were not considered in subsequent analysis.

### 2.3. Data Analysis

#### 2.3.1. Phylogenetic Reconstruction and WGS-Based Cluster Inference

On the basis of concatenated sequence data, the most suitable substitution model for the dataset was implemented using the maximum likelihood ratio test (Jmodel test) [25]. A maximum likelihood tree was calculated based on the concatenated sequence alignment using FastTree [26] with a general time-reversible (GTR) substitution model (best according to Jmodel test), 1000 resamples, and γ 20 likelihood optimization to account for the rate of heterogeneity among sites. The phylogenetic tree was inspected and rooted with FigTree software (Institute of Evolutionary Biology, University of Edinburgh, Edinburgh, Scotland). Graphical presentation was performed with Evolview [27]. The concatenated sequence data were also used to calculate maximum parsimony trees using BioNumerics software version 7.6 (Applied Maths, BioMérieux, Sint-Martens-Latem, Belgium). Genome clusters were defined by grouping all sequenced isolates within a maximum distance of ≤12 and ≤5 SNPs between one strain and the neighboring strain [28,29].

#### 2.3.2. Molecular Drug Resistance Prediction

Genes associated with drug resistance mechanisms were individually investigated and not considered for phylogenetic approaches [24].

The SNP dataset was examined for known resistance-conferring mutations against first-line and second-line antibiotics [24]. In brief, for rifampicin (RIF) resistance, mutations that confer resistance on the *rpoB* gene were analyzed. Other RNA-polymerase genes, i.e., *rpoA* and *rpoC*, were analyzed with regard to putative compensatory mutations. We also investigated *katG*, *inhA*, *ndh*, and *fabG1* genes for mutations that confer resistance to isoniazid (INH). We screened *pncA* and *rpsA* genes for mutations conferring resistance to pyrazinamide (PZA), and *embB*, *embC*, and *embA* genes for mutations associated with ethambutol (EMB) resistance. In the case of streptomycin (STR) resistance, the *rpsL*, *rrs*, and *gidB* genes were investigated for mutations conferring resistance. For isolates that were MDR (i.e., resistance towards INH and RIF), genes conferring resistance against second-line drugs were examined. The *gyrA* and *gyrB* genes were investigated for resistance against fluoroquinolones (FQs), while the *rrs* gene was studied for resistance against aminoglycosides. In addition, *tlyA* and *thyA* genes were analyzed for capreomycin (CAP) and para-aminosalicylic acid (PAS) resistance, respectively. For ethionamide (Eto) resistance, *ethA* and *fabG1* genes were studied [24].

#### 2.3.3. Statistics 

Demographic data were entered, cleaned, and analyzed using the SPSS statistical software package, version 20 (SPSS Inc., Chicago, IL, USA). Chi-square test and Fisher exact test were used to compare categorical variables. All variables were tested using logistic regression analysis. A *p*-value ≤ 0.05 was considered significant.

### 2.4. Ethical Considerations

This study was approved by the Scientific and Ethical Review Unit of Kenya Medical Research Institute (SSC Protocol No. 2830). All patients recruited into the study were required to sign a written informed consent for collection and analysis of samples and demographic and clinical data. If the patient was illiterate or less than 18 years old, a caretaker was requested for consent. All the information collected from cases was kept confidential. All patient identifiers were removed prior to data analysis. During the study, any diagnosed TB cases (i.e., smear-positive) were registered at the health centers and treated according to the National Tuberculosis and Leprosy Programme (NTLP) guidelines [30].

The NTLP guidelines state that all patients who have not been on TB therapy previously should have a two-month initial phase of treatment consisting of INH, RIF, PZA, and EMB followed by a continuation phase of EMB and INH for six months or INH and RIF for four months [30]. For retreatment TB patients, the intensive phase is three months of daily injections of STR and swallowing of INH, RIF, PZA, and EMB, followed by five months of continuation phase with INH, RIF, and EMB [30].

## 3. Results

### 3.1. Study Population

A total of 479 sputum samples were collected from smear-positive patients in health centers in Nairobi and its surroundings (Appendix A). Based on hospital records, the recruited patients comprised 10–20% of all patients who received a diagnosis of TB during the study period. 

After excluding individuals with missing data, we found that the overall mean age of the investigated patients was 31.7 ± 8.3 years. Moreover, we observed that most of the enrolled individuals (64.4%, 246/382) were males and were newly diagnosed TB cases (85.3%, 326/382). Most of the study cases were from health centers located in non-slum areas (i.e., urban or developed areas with well-constructed houses, commercial buildings, roads, and bridges) (55.8%, 215/385), while 44.2% (170/385) were from health centers located in slums (informal settlements) (Appendix A).

Out of all collected sputum specimens, 48/479 (10.0%) were either culture-negative or contaminated (Figure 1). Furthermore, 31 isolates were subsequently excluded from the analysis due to quality issues or the identification of more than one strain (*n* = 15) in the DNA isolate. The final analysis comprised 385/479 (80.4%) WGS datasets, each representing one isolate per patient (Figure 1).

Comparison of the two patient cohorts (i.e., 2010 cohort vs. 2014–2015 cohort) revealed significant differences between the proportions of drug-susceptible and drug-resistant strains (X^2^ = 25.3, *p*-value ≤ 0.001), the proportions of clustered and not clustered strains based on ≤5 SNP threshold (X^2^ = 14.2, *p*-value ≤ 0.001), and the proportions of male and female sex (X^2^ = 4.97, *p*-value = 0.026) after excluding individuals with missing information from the analysis (Table 1). Furthermore, the proportions of patients residing in urban and slum living conditions differed significantly between the two cohorts (X^2^ = 178.2, *p*-value ≤ 0.001). While in the 2010 cohort 91.7% (166/181) of the patients came from urban areas, only 24.0% (49/204) of the patients from the 2014–2015 cohort were living in an urban setting (Table 1).

### 3.2. MTBC Population Structure

We performed WGS successfully on 400 MTBC strains. We excluded 15 samples/datasets from the final analysis because of the identification of more than one strain and built a maximum likelihood phylogeny upon a concatenated sequence alignment comprising 18,167 SNPs to investigate the MTBC population structure using 385 strains (Figure 2). Phylogenetic lineages (Ls) were inferred from canonical SNPs specific for certain MTBC sublineages based on a recently introduced SNP barcode classification [31,32]. Strains of L4 (Euro-American super-lineage, *n* = 215/385, 55.8%) were predominant, followed by strains of L3 (Delhi/CAS, *n* = 99/385, 25.7%), strains of L2 (Beijing, *n*= 56/385, 14.5%), and strains of L1 (East African Indian, *n* = 14/385, 3.6%). There was one *M. bovis* strain identified (Figure 2).

Strains of L4 (the Euro-American super-lineage) were further classified into several sublineages as described previously [31,32]. Strains of the Latin American Mediterranean (LAM) genotype (*n* = 70/215, 32.5%) comprised the MTBC sublineages 4.3.2, 4.3.4.1, 4.3.4.2, and 4.3.4.2.1 and were most frequent, followed by S-type (4.4.1.1, *n* = 55/215, 25.5%), Turkish (TUR) (4.2.2, *n* = 27/215, 12.6 %), and Haarlem (4.1.2 and 4.1.2.1, *n* = 18/215, 8.4%) (Appendix A). Other sublineages identified include Kenya H37Rv-like (4.8, *n* = 13/215, 6%), Uganda I (4.6.1.1, *n* = 7/215, 3.2%), Uganda II (4.6.1.2 *n* = 10/215, 4.6%), X-type (4.1.1.1, *n* = 7/215, 3.2%), Cameroon (4.6.2 *n* = 6/215, 2.8%), Ghana (4.1 *n* = 1/215, 0.5%), and undefined Euro-American (4.0, *n* = 1/215, 0.5%) (Appendix A).

### 3.3. Genome-Based Drug Resistance Prediction

Overall, 37 out of the 385 (9.6%) MTBC isolates investigated were found to be resistant to at least one of the first-line drugs, of which 20 (5.2%) were MDR. Monodrug resistance to INH, RIF, or PZA was observed in 15 (3.9%), 1 (0.3%), and 1 (0.3%) of the MTBC strains, respectively (Appendix A).

A detailed analysis of variants in resistance-associated genes revealed that 9.1% (35/385) of the strains had mutations conferring resistance to INH, with the Ser315Thr in the *katG* gene being the dominant observed mutation (60.0%, 21/35) (Table 2). The -15 c/t mutation in the *fabG-inhA* promoter region was the second most dominant INH resistance determinant (22.8%, 8/35). Variants in the *rpoB* gene that confer resistance to RIF were found in 21 (5.5%) strains, with the Ser450Leu mutation being the most common (52.3%, 11/21) (Table 2). Moreover, mutations at codon 445 leading to His445Tyr, His445Arg, and His445Asp amino acid substitutions were observed in seven (33.3%) strains.

The most common mutation mediating STR resistances was the Lys43Arg (69.2%, 9/13) at the *rpsL* gene, followed by the 513 a/c (23.0%, 3/13) mutation in the *rrs* gene. Eleven strains were resistant to EMB, with Met306Ile being the frequent mutation observed in the *embB* gene (63.6%, 7/11) (Table 2). The *pncA* gene was analyzed for mutations conferring resistance to PZA. The Lys96Thr mutation was found to be the most common (50.0%, 6/12).

Only one strain was found with a fluoroquinolone resistance mediating mutation in the *gyrA* gene (Asp94Gly, 1/385, 0.3%). This strain was hence defined as pre-extensively drug-resistant (XDR) which is an MDR and resistant to any fluoroquinolone. There were no resistance markers identified for other second-line anti-TB drugs.

When we compared the proportions of drug-resistant (i.e., any antibiotic resistance) and wild-type (fully susceptible) strains between sex groups, age groups, and TB history, we found no differences between male and female, ≤30 and >30 age groups, and new and retreatment TB cases with a *p*-value of ≥0.247 (Table 3). However, the proportion of infection with a drug-resistant strain was observed to be significantly associated with urban living conditions (OR = 3.8, 95% CI 1.62–8.83, *p*-value = 0.002) (Table 3). Notably, all MDR strains detected in this study were found in urban areas.

Overall, 30% (6/20) of the MDR MTBC strains were identified with molecular resistant markers to all first-line drugs (Appendix A). Furthermore, L3 (7/20, 35.0%) and L4 (7/20, 35.0%) equally dominated among the MDR TB strains. Comparison of the proportions of drug-resistant strains across the phylogenetic lineages (L1–L4) using logistic regression revealed no statistical differences (Table 3).

### 3.4. Genome-Based Cluster Analysis

We performed a genome-based cluster analysis with thresholds of a maximum genetic distance of ≤12 and ≤5 SNPs between any two MTBC strains to obtain an indication of potential TB transmission events [28,29]. Overall, 56.6% (218/385, 95% CI 51.6–61.5) of the MTBC strains were grouped into 61 genome clusters based on a ≤12 SNP basis. The cluster size ranged between 2 and 28 strains/patients per cluster (Appendix A). If a stricter genetic distance of ≤5 SNPs was used for cluster analysis, 30.4% (117/385, 95% CI 26.0–35.2) of the strains were grouped into 41 clusters ranging from 2 to 21 strains/patients (Appendix A).

Stratifying by phylogenetic MTBC lineage showed that L2 strains had the highest genome clustering rate using both ≤12 and ≤5 SNP distance (66.1% and 35,7%), followed by L4 (57.2% and 34,4%). L2 and L4 strains further had a higher odds ratio to be in a molecular cluster (L2, OR 2.1, *p*-value= 0.051 and L4 OR 1.9, *p*-value = 0.019) using a SNP threshold of ≤5 (Table 4). This indicates that L2 and L4 strains are more effectively transmitted as compared to L3 strains. However, a statistical difference was only observed for the strict threshold of ≤5 SNP genetic distance (Table 4).

Among the strains of the L4 MTBC sublineages, strains of the L4.4.1.1 (S-type) sublineage formed the largest genome cluster comprising 28 strains/patients. Overall, L4.4.1.1 (S-type) (OR 2.19, *p* = 0.001), L4.1.1.1 (X-type) (OR 2.7, *p* = 0.028), and L4.8 (Kenya H37Rv-like) (OR 2.12, *p* = 0.017) were significantly associated with molecular clusters. 

### 3.5. Transmission of MDR MTBC Strains

Based on ≤12 and ≤5 SNP genetic distances, 60% (12/20, three clusters) and 50% (10/20, three clusters) of MDR strains were clustered, respectively (Appendix A). Within the genomic clusters, MDR strains showed identical resistance-conferring mutations in the *katG* gene. In the *rpoB* gene, most strains (11/20, 55.0%) carried the *rpoB* mutation Ser450Leu as well as *katG* Ser315Thr (18/20, 90%). The largest MDR cluster (with both thresholds, ≤12 and ≤5 SNPs) comprised six L4.4.1.1 (S-type) strains that all shared the following resistance-conferring mutations: *katG* Ser315Thr, *rpoB* Ser450Leu, *pncA* Lys96Thr, *rpsL* Lsy43Arg, and *embB* Met306Ile. One of the strains of this cluster already evolved to pre-XDR. All patients within this MDR cluster were from urban living areas in Nairobi.

## 4. Discussion

In this study, we elucidated the MTBC population structure and transmission events in Nairobi, Kenya, by using WGS. We showed that pulmonary TB is dominantly caused by strains of L4 and L3 and noted a substantial cluster rate which points towards ongoing transmission of MTBC strains. Moreover, all patients infected with MDR strains were from urban areas of Nairobi. We further found a cluster that consists of strains that are resistant to all first-line anti-TB drugs, with one of them already evolved to pre-XDR. These findings emphasize the need for more focused infection control measures and contact tracing of patients with MDR TB to break the transmission chains.

We revealed that the MTBC population comprises strains of four lineages in Nairobi, namely L1–L4 and one *M. bovis* strain. Strains of L4 and L3 are the main causative agents of TB comprising 55.8% and 25.7% of the investigated strains, respectively. These findings mirror reports from neighboring countries, such as Ethiopia [33], Tanzania [34], and Uganda [35] but are in contrast to the findings from Shuaib et al. [12] and Ejo et al. [36] in Eastern Sudan and Northwest Ethiopia where L3 strains were the main etiological agent of TB.

We found that strains of the LAM (L4.3.1, L4.3.3, and L4.3.4) sublineages are the most prevalent in Nairobi in comparison with strains of other L4 sublineages. Stucki et al. [37] have identified strains of certain sublineages of L4, such as LAM (L4.3), Haarlem (L4.1.2), and PGG3 (L4.10), as generalists as these strains can cause TB in all host genetic backgrounds globally. Therefore, the high prevalence of the strains of the generalist LAM sublineage might explain the dominance of L4 strains in Nairobi in Kenya. Another possible explanation for the success of L4 might be related to the observation that some L4 strains are capable of evading the host immune response and rapidly progressing to TB disease and thus are more often transmitted [15,36].

As mentioned before, L3 strains were the second most prevalent causative agent of TB in Nairobi. It’s been previously suggested that L3 strains have an evolutionary origin in South Asia. However, L3 strains have frequently been isolated from TB patients in East and North Africa [12,34,36,38], and one could speculate if these strains have co-evolved with and adapted to their East African hosts and consequently developed specific biological/phenotypic traits in this particular host population [12]. The high prevalence of L3 strains in Kenya could also be related to recent and/or past movements of people from the Indian subcontinent (i.e., migration, tourism, and trade).

In this study, we found that L2 strains comprised 14.5% of the investigated strains, which is higher than former reports where the prevalence of L2 strains was 8.4% in 2004 [39]. This upward trend indicates that strains of this lineage are also successfully transmitted in the population. Moreover, the reported proportion of L2 strains in the present study is higher than that in other countries of the region, including Sudan (0.6%) [12], Tanzania (4.1%) [34], Uganda (1.2%) [38], and Ethiopia (0.5%) [40]. It is known that L2 strains have their origin in East Asia, where they are responsible for multiple TB epidemics [10,41]. The expansion of L2 strains on the African continent has been suggested to be associated with increased transmission, high virulence, and a rapid progression to disease, rather than with drug resistance [10,41]. In addition, labor migration has been discussed as a possible factor explaining the increasing prevalence of L2 strains, mainly in East African countries [41].

We found evidence of direct patient-to-patient MTBC transmission as one-third (i.e., 30.4%) of the isolates/patients were associated with genomic clusters using a ≤5 SNP threshold in this study. Surprisingly, the transmission of TB is significantly higher in urban areas than in slum areas of Nairobi using both ≤12 and ≤5 SNP distances (34.9% in urban areas vs. 24.7% in slum areas, Exp(B) = 1.6, *p*-value = 0.032). This indicates that a higher risk of being infected due to transmission is related to a certain hotspot. Comparing the cluster rate across all phylogenetic lineages showed that L2 and L4 strains had an increased odds ratio of being in a molecular cluster (d5: L2, OR 2.1, *p*-value = 0.051 and L4 OR 1.9, *p*-value = 0.019). Additionally, among the strains of L4 MTBC sublineages, strains of the S-type, X-type, and Kenya H37Rv-like are likely more successful, i.e., have a higher odds to be in a molecular cluster. The observed cluster rate in Nairobi is comparable to the cluster rates reported in other high TB burden countries, such as Sudan [12], South Africa [16], and Ethiopia [42]. 

Early detection of resistance to anti-TB drugs is important for successful treatment and control of MDR TB [43]. In this study, 9.6% of the strains were found to be resistant to at least one first-line anti-TB drug, with 5.2% being MDR strains. This proportion of MDR strains is higher than those previously reported in Kenya [44,45] and in other countries, e.g., Uganda (1.7%) [46]. Nevertheless, in Sudan (9.0%) [12] and Tanzania (6.3%) [47], higher proportions of MDR strains have been reported. The variations in resistance rates observed in our study and the previous studies in Kenya could be due to the study design and sample size obtained [44,45]. Information on susceptibility patterns of strains against anti-TB drugs is essential for the control and surveillance of TB. Therefore, early diagnosis and treatment, improving treatment outcomes, and expanding diagnostic capacity for mycobacterial culture and drug susceptibility testing are crucial to limit the spread of drug resistant MTBC strains, especially MDR. In the case of the MDR and RR TB cases, rapid detection of resistance, e.g., by GeneXpert, followed by extended phenotypic drug susceptibility testing is essential for patient isolation and establishment of second-line anti-TB therapy. Otherwise, prolonged periods of ineffective treatment will likely allow continuous transmission of MDR cases with disastrous consequences for TB control in the future.

Furthermore, we found a high cluster rate among MDR strains (60% based on ≤12 SNP threshold and 50% based on ≤5 SNP threshold) in combination with an increased risk of infections with drug-resistant strains in urban living areas in Nairobi. This is in line with a recent Kenya TB prevalence survey that indicated that the majority of TB cases are found in urban areas [14]. Therefore, successful control of TB in these settings should focus on addressing the influence of associated social and economic factors as well as strengthen TB control measures including contact tracing, early TB case detection, and adherence to treatment.

One limitation of our study is the lack of comprehensive epidemiological and clinical data, such as HIV status, diabetes, and other comorbidities, as well as treatment outcome. This limited a detailed analysis of epidemiological and clinical factors associated with recent transmission of MTBC strains (i.e., by using number of clustered strains as surrogate). Another limitation is the sampling bias of the two cohorts. Higher proportions of MDR and clustered strains were collected mainly from TB patients living in urban areas in the 2010 study cohort. This could have biased the association between any drug resistance and urban settings in the combined analysis.

## 5. Conclusions

Our study demonstrates that TB is predominantly caused by L4 strains in the urban and slum settings of Nairobi, Kenya. WGS analysis provides a better understanding of transmission dynamics of MTBC strains and their molecular drug resistance determinants. We found a cluster of strains that are resistant to all first-line anti-TB drugs. One of the strains of this molecular cluster has already evolved to a pre-XDR genotype via the acquisition of FQ resistance. Importantly, we observed that all MDR strains are associated with urban living environments and half of them are linked to direct patient-to-patient transmission, highlighting the urgent need to increase efforts to identify MDR TB cases and trace patient contacts to contain the spread of the disease in Nairobi. Adoption of the use of WGS may further allow for improved disease surveillance in high TB burden countries. Moreover, additional facilities in which molecular diagnosis of TB and drug susceptibility testing can be performed are required.

## Figures and Tables

**Figure 1 genes-13-00475-f001:**
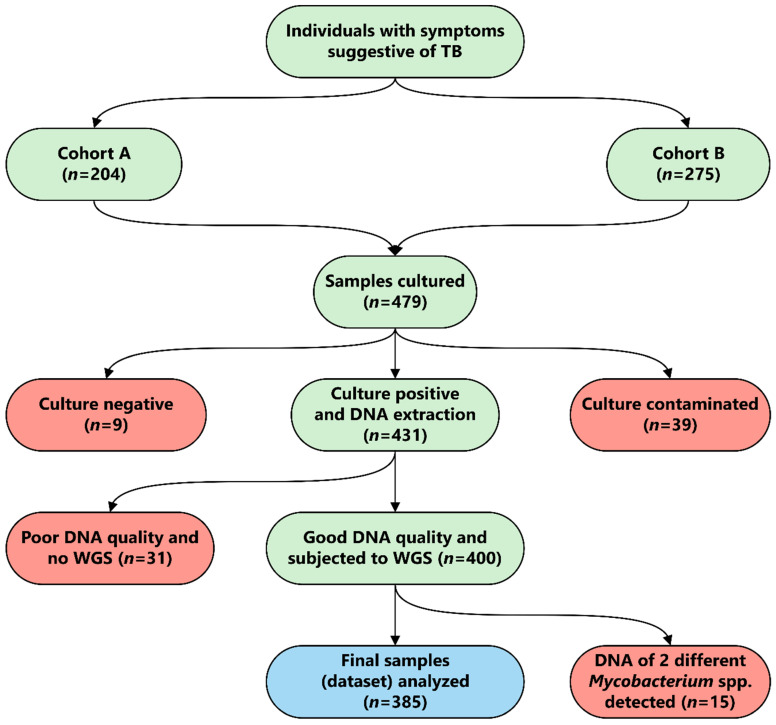
Workflow for the study “Molecular epidemiology of *Mycobacterium tuberculosis* complex strains in urban and slum settings of Nairobi, Kenya”. TB = tuberculosis, Cohort A = patients enrolled in 2010, Cohort B = patients enrolled in 2014–2015, WGS = whole-genome sequencing, spp. = species.

**Figure 2 genes-13-00475-f002:**
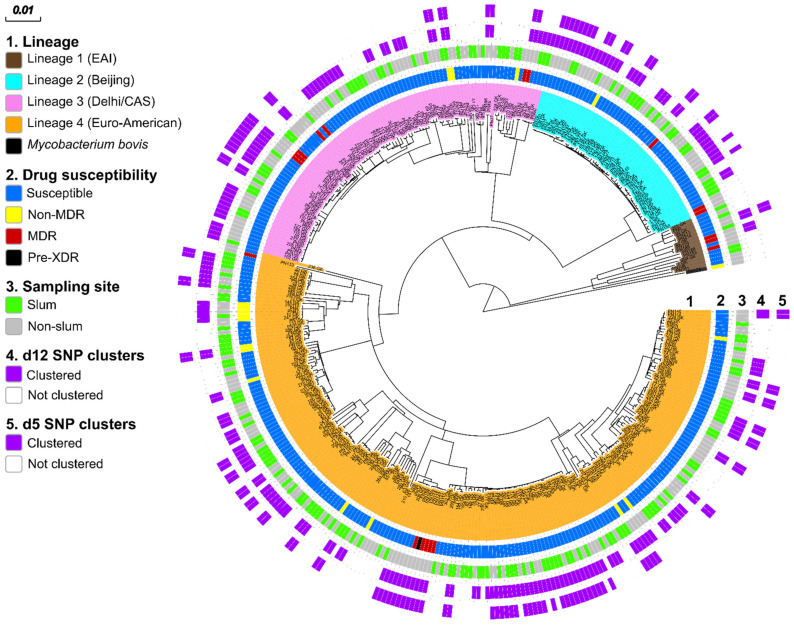
MTBC population structure in urban and slum areas of Nairobi, Kenya. Maximum likelihood tree based on 18,167 concatenated single-nucleotide polymorphisms (SNPs) using a general time-reversible substitution model. Colored bars code for (inner to outer ring) *M. bovis* and MTBC lineages (L1–4); genotypic DST results stratified to MDR, non-MDR, and fully susceptible; sampling location; and clustered and non-clustered strains (SNP distance ≤12 and ≤5). DST = drug susceptibility testing, MDR = multidrug-resistant (simultaneous resistance towards isoniazid and rifampicin); Pre-XDR= pre-extensively drug-resistant (an MDR which is also resistant to any fluoroquinolone), MTBC = *Mycobacterium tuberculosis* complex.

**Table 1 genes-13-00475-t001:** Comparison of the two patient cohorts investigated by using chi-square test in Nairobi, Kenya (2010–2015).

Variable	No.	Cohort A	Cohort B	X^2^	*p*-Value
**Lineage**				7.70	0.103
L1 (EAI)	14	9 (5.0%)	5 (2.5%)		
L2 (Beijing)	56	30 (16.6%)	26 (12.7%)		
L3 (Delhi/CAS)	99	52 (28.7%)	47 (23.0%)		
L4 (Euro-American)	215	89 (49.2%)	126 (61.8%)		
*M. bovis*	1	1 (0.5%)	0 (0.00%)		
**DST**				25.3	<0.001 ^a^
Susceptible	348	151 (83.4%)	197 (96.6%)		
MDR	20	20 (11.1%)	0 (0.00%)		
Non-MDR	17	10 (5.5%)	7 (3.4%)		
**d12**				3.08	0.079
Clustered	218	111 (61.3%)	107 (52.5%)		
Not clustered	167	70 (38.7%)	97 (47.5%)		
**d5**				14.2	<0.001 ^a^
Clustered	117	72 (39.8%)	45 (22.1%)		
Not clustered	268	109 (60.2%)	159 (77.9%)		
**TB history**				0.11	0.747
New	326	152 (84.0%)	174 (85.3%)		
Retreatment	56	28 (15.5%)	28 (15.5%)		
NA ***	3	1 (0.5%)	2 (1.0%)		
**Age group**				0.11	0.917
≤30	166	63 (34.8%)	103 (50.5%)		
>30	161	62 (34.3%)	99 (48.5%)		
NA ***	58	56 (30.9%)	2 (1.0%)		
**Sex**				4.97	0.026 ^a^
Male	246	105 (58.0%)	141 (69.1%)		
Female	136	75 (41.4%)	61 (29.9%)		
NA ***	3	1 (0.6%)	2 (1.0%)		
**Sampling area**				178.2	<0.001 ^a^
Urban	215	166 (91.7%)	49 (24.0%)		
Slum	170	15 (8.3%)	155 (76.0%)		
**Total**	**385**	181 (47.0%)	204 (53.0%)	**─**	**─**

No. = number, A = cohort 2010, B = cohort 2014–2015, DST = drug susceptibility testing, MDR = simultaneous resistance towards isoniazid and rifampicin, non-MDR = resistance towards drugs other than isoniazid and rifampicin, d12 = genetic distance of ≤12 single-nucleotide polymorphisms (SNPs), d5 = genetic distance of ≤5 SNPs, NA = not available, * = variable (individuals with missing data) excluded from analysis, ^a^ = variable with a chi-square *p*-value of ≤ 0.05.

**Table 2 genes-13-00475-t002:** Detected mutations that mediate resistance to first-line anti-TB drugs in 37 resistant MTBC strains in urban and slum areas of Nairobi, Kenya (2010–2015).

Drug	Gene	Mutation	Codon Change	Total
INH	*katG*	Ser315Thr	agc/aCc	21
		Ser315Arg	agc/aGa	2
	*fabG1-inhA*	−15 c/t	C→T	8
	*inhA*	Leu203Leu	ctg/ctA	2
		Ile194Thr	atc/aCc	2
RIF	*rpoB*	Gln432Pro	Caa/cCa	2
		His 445Tyr	Cac/Tac	5
		His 445Arg	Cac/cGc	1
		His445Asp	Cac/Gac	1
		Ser450Leu	Tcg/tTg	11
		Leu452Pro	Ctg/cCg	1
STR	*rpsL*	Lys43Arg	Aag/aGg	9
	*gidB*	Ala138Val	Gcg/gTg	1
	*rrs*	513 a/c	A→C	3
EMB	*embB*	Met306Ile	Atg/atA	7
		Met306Val	Atg/Gtg	2
		Asp328Gly	Gat/gGt	1
		Asp354Ala	Gac/Aac	1
PZA	*pncA*	Gln10Pro	cag/cCg	1
		His57Asp	cac/Gac	1
		Asp63Gly	gac/gGc	2
		Lsy96Thr	aag/aCg	6
		Thr135Pro	acc/Ccc	1
		Gln141_	cag/tag	1
FQs	*gyrA*	Asp94Gly	gac/Cac	1

INH = isoniazid, RIF = rifampicin, STR, streptomycin, EMB = ethambutol, PZA = pyrazinamide, FQs = fluoroquinolones.

**Table 3 genes-13-00475-t003:** Comparison of the proportions of drug-susceptible and drug-resistant strains across the sociodemographic variables and MTBC phylogenetic lineages by using logistic regression in Nairobi, Kenya (2010–2015).

Variables	No.	No. of Wild Type	No. of Resistant (%)	OR (95% CI)	*p*-Value
**Gender**					
Male	246	226	20 (8.10)	Ref	
Female	136	120	16 (11.8)	1.5 (0.75–3.01)	0.247
NA *	3	2	1 (33.3)	─	─
**Age**					
≤30	166	148	18 (10.8)	Ref	
>30	161	149	12 (5.70)	1.5 (0.70–3.24)	0.291
NA *	58	51	7 (12.1)	─	─
**Study area**					
Slum	170	163	7 (4.1)	Ref	
Urban	215	185	30 (14.0)	3.8 (1.62–8.83)	0.002
**TB history**					
New	326	296	30 (9.2)	Ref	
Retreatment	56	50	6 (10.7)	1.2 (0.47–2.99)	0.721
NA *	3	2	1 (33.3)	─	─
**Lineage**					
L2 (Beijing)	56	52	4 (7.1)	Ref	
L1 (EAI)	14	11	3 (21.4)	3.5 (0.69–18.1)	0.129
L3 (Delhi/CAS)	99	89	10 (10.1)	1.5 (0.44–4.89)	0.539
L4 (Euro-American)	215	196	19 (8.8)	1.3 (0.41–3.86)	0.868
*M. bovis* *	1	0	1 (100)	─	─
**Total**	**385**	**348**	**37 (9.6)**	**─**	**─**

No. = number, OR = odds ratio, CI = confidence interval, NA = not available, L = lineage, * = variable (individuals with missing data) excluded from the regression model.

**Table 4 genes-13-00475-t004:** Comparison of the proportions of clustered and not clustered strains across the detected MTBC phylogenetic lineages by using logistic regression in Nairobi, Kenya (2010–2015).

Variable	No.	d12				d5			
		NC	C (%)	OR (95% CI)	*p*-Value	NC	C (%)	OR (95% CI)	*p*-Value
**Lineage**									
L3 (Delhi/CAS)	99	45	54 (54.5%)	Ref		78	21 (21.2%)	Ref	
L2 (Beijing)	56	19	37 (66.1%)	1.6 (0.82–3.20)	0.163	36	20 (35.7%)	2.1 (1.00–4.28	0.051
L4 (Euro-American)	215	92	123 (57.2%)	1.1 (0.69–1.80)	0.658	141	74 (34.4%)	1.9 (1.12–3.41)	0.019
L1 (EAI) *	14	10	4 (28.6%)	─	─	12	2 (14.3%)	─	─
*M. bovis* *	1	0	0 (0.00%)	─	─	0	0 (0.00%)	─	─
**Total**	**385**	**166**	**218 (56.6)**	**─**	**─**	**267**	**117 (30.5)**	**─**	**─**

No. = number, d12 = genetic distance of ≤12 single-nucleotide polymorphisms (SNPs), d5 = genetic distance ≤ 5 SNPs, NC = number of not clustered, C = number of clustered, OR = odds ratio, CI = confidence interval, NA = not available, * = variable excluded from the regression model.

## Data Availability

Generated raw sequencing reads in this study were submitted to the European Nucleotide Archive (ENA) under the project accession number (PRJEB50767).

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
