# Peer review of "Molecular Epidemiology of Mycobacterium tuberculosis Complex Strains in Urban and Slum Settings of Nairobi, Kenya"

_genes, 2022, doi:10.3390/genes13030475_

Round 1

Reviewer 1 Report

This paper presents the results of whole genome sequencing of around 400 mycobacterium tuberculosis isolates from Nairobi, Kenya.

The study is interesting and well designed.

Comments

  1. I could not access the supplementary file describing the setting from where the samples were taken.
  2. A description of the demographics and definition of a slum area and urban area would be helpful
  3. More information on the index cases would be useful - this is mentioned as a limitation by the authors
  4. Was DST carried out for the samples and if so was this compared to the genetic predictions?

Reviewer 2 Report

TB, and MDR TB in paticular, is still a serious problem in Kenya.
Investigations performed worldwide show that WGS for determining mutations associated with MDR TB gives quick results of MBT resistance to key anti-TB drugs, which allows the doctor to prescribe an adequate chemotherapy regimen to the patient at the very beginning of treatment. However, upon receipt of the MTB culture, the genotypic method
should be further duplicated by the phenotypic drug sensitivity testing
method, based on the results of which, if necessary, the treatment regimen
is adjusted. Here the authors used TB-WGS for drug resistance prediction and epidemiological services in Nairobi. 

  1. The authors did not mention any ethical approvement of the study. Did probands sign informed consent?
  2. Also the description lacks information on the type of chemothrapy the patients received.
  3. At wich interval from the start of chemotherapy the probes for MTB cultures were obtained?

Reviewer 3 Report

Dear Editor,

Thank you for the invitation.

After careful reading, the manuscript is very interesting and there are a few comments for further improvement of the manuscript.

Sincerely,
